# High Incidence of Respiratory Syncytial Virus in Children with Community-Acquired Pneumonia from a City in the Brazilian Pre-Amazon Region

**DOI:** 10.3390/v15061306

**Published:** 2023-05-31

**Authors:** Valéria Fontes, Hivylla Ferreira, Marilene Ribeiro, Aruanã Pinheiro, Carlos Maramaldo, Eduardo Pereira, Luís Batista, Antonio Júnior, Luis Lobato, Fabiano Silva, Luis Sousa, Washington Lima, Claudia Lima, Suzany Soczek, Rafael Carvalho, Mirleide Santos, Elizabeth Fernandes, Eduardo Sousa, Lidio Neto

**Affiliations:** 1Laboratory of Virology, Post-Graduate Programme in Microbial Biology, CEUMA University, São Luís, MA 65075-120, Brazil; 2Central Public Health Laboratory of Maranhão—LACEN-MA, Osvaldo Cruz Institute, São Luís, MA 65020-904, Brazil; 3Hospital of Federal University of Maranhão, HU-UFMA, São Luís, MA 65020-070, Brazil; 4Postdoctoral Program in Microbial Biology, CEUMA University, São Luís, MA 65075-120, Brazil; 5Postgraduate Program in Adult Health (PPGSAD), Federal University of Maranhão, UFMA, São Luís, MA 65080-805, Brazil; 6Post-Graduate Programme in Tropical Medicine, FIOCRUZ-RJ, Rio de Janeiro, RJ 21040-360, Brazil; 7Post-Graduate Programme in Biodiversity and Biotechnology (BIONORTE), CEUMA University, São Luís, MA 65075-120, Brazil; 8Post-Graduate Programme in Biotechnology Applied to Child and Adolescent Health, Pelé Pequeno Príncipe Research Institute, Curitiba, PR 80250-060, Brazil; 9Evandro Chagas Institute, Ananindeua, PA 67030-000, Brazil

**Keywords:** respiratory tract infections, clinical epidemiology, child

## Abstract

Introduction: Although fewer children have been affected by the severe form of the coronavirus disease 2019 (COVID-19), community-acquired pneumonia (CAP) continues to be the leading global cause of child hospitalizations and deaths. Aim: This study investigated the incidence of respiratory syncytial virus (RSV) as well its subtypes (RSV A and B), adenovirus (ADV), rhinovirus (HRV), metapneumovirus (HMPV), coronavirus (NL63, OC43, 229E and HKU1), parainfluenza virus subtypes (PI1, PI2 and PI3), bocavirus and influenza A and B viruses (FluA and FluB) in children diagnosed with CAP during the COVID-19 pandemic. Methods: A total of 200 children with clinically confirmed CAP were initially recruited, of whom 107 had negative qPCR results for SARS-CoV-2 and were included in this study. Viral subtypes were identified using a real-time polymerase chain reaction in the nasopharyngeal swab samples. Results: Viruses were identified in 69.2% of the patients. RSV infections were the most frequently identified (65.4%), with type RSV B being the most prevalent (63.5%). In addition, HCoV 229E and HRV were detected in 6.5% and 3.7% of the patients, respectively. RSV type B was associated with severe acute respiratory infection (ARI) and a younger age (less than 24 months). Conclusions: New strategies for preventing and treating viral respiratory infections, particularly RSV infections, are necessary.

## 1. Introduction

Respiratory infections, especially acute ones, have always had significant relevance for health services, even before the COVID-19 pandemic, largely because they are always associated with increased hospitalizations, morbidity and mortality, especially among children. One of the great challenges related to respiratory infections is the fact that their clinical presentation is not exclusive to the virus causing the infection; therefore, a molecular diagnosis consisting of a viral panel ends up being fundamental for the identification of the virus and to make appropriate therapeutic decisions [1].

Infections of the lower respiratory tract are the major cause of death in children worldwide [2,3,4,5], accounting for more than one million deaths annually [6]. A high number of deaths among children under 5 years of age is observed, mainly in developing countries, including Brazil. Since the beginning of the coronavirus disease 2019 (COVID-19) pandemic, more than six million deaths have been recorded worldwide. The severe form of COVID-19 has affected fewer children in comparison to adults; however, community-acquired pneumonia (CAP) continues to be the leading cause of child hospitalizations and deaths worldwide.

Respiratory infections, in general, can range from asymptomatic or mild clinical conditions, such as flu-like syndrome (SG), to more intense presentations that can result in severe acute respiratory syndrome (SARS), causing hospitalization and sometimes the need for ventilator support, with a high risk of death [7].

Given the complexity and diversity of the etiological agents of CAP, a quick and accurate diagnosis becomes extremely important, as it allows for safer, more effective treatment and avoids possible complications of the disease, especially those due to viral infections, and avoids the use of medicines by mistake. In this context, among the main pathogenic agents of CAP, respiratory syncytial virus (RSV), human rhinovirus (HRV), adenovirus (ADV) and the influenza viruses, as well as their subtypes (FluA and FluB), stand out [8].

In a recent study published by our group, we identified rhinovirus as the most frequent virus found among 150 children admitted for hospitalization with CAP. In that study, RSV and FluA infections were associated with severe CAP [9]. Therefore, here we aimed to investigate the incidence of RSV as well as its subtypes (RSV A and B), ADV, HRV, HMPV, coronavirus (NL63, OC43, 229E and HKU1), parainfluenza virus subtypes (PI1, PI2 and PI3), bocavirus and influenza A and B (FluA and FluB) viruses in children with ARI. The contributing risk factors for clinically confirmed CAP in children during the COVID-19 pandemic were also analyzed.

## 2. Materials and Methods

### 2.1. Study Population

The study was conducted at the Dr. Odorico de Amaral Matos Children’s Hospital and the Dr. Juvencio Mattos Children’s Hospital, both academic and public hospitals serving Sao Luis, a city located in the Brazilian pre-Amazon region, from January 2021 to March 2021.

The study population was composed of 200 consecutive patients ranging from 1 month to 10 years of age, who were invited to enroll in the study at the time of hospital admission, according to protocol definitions and inclusion and exclusion criteria. Patients were selected by the clinic dedicated to lung infections. Of those, ninety-three tested positive for SARS-CoV-2, detected by RT-qPCR, and were excluded from the study. The remaining 107 patients were negative for the virus and were included in the study (Figure 1).

Pneumonia cases were defined according to the guidelines of the Brazilian Society of Pulmonology and Physiology [10]. Through the use of imaging tests such as chest X-rays, the presence of pulmonary consolidation (a dense or fluffy opacity), pulmonary infiltrate (alveolar or interstitial densities) or pleural effusion was observed. In addition, the presence of two or more symptoms of an acute lower respiratory tract illness were observed, such as: cough, fever, difficulty breathing, age-adjusted tachypnea (≥50 breaths/min for children aged 2 to 11 months, ≥40 breaths/min for children aged ≥12 months) and/or wheezing. Severe pneumonia (severe CAP) was defined as the presence of lower thoracic concavity, an inability to eat or drink, vomiting, convulsions, lethargy, unconsciousness, severe malnutrition, oxygen saturation (SpO2) below 90% using pulse oximetry and/or central cyanosis.

The exclusion criteria were: (a) undergoing otolaryngologic surgery or (b) the presence of a compromising chronic debilitating disease (anatomic abnormalities of the respiratory tract, cancer, chronic pulmonary illness or immunological defects with clinical repercussions).

### 2.2. Ethical Approval and Consent to Participate

This study was approved by the Research Ethics Committee of CEUMA University, under number CAAE No. 467.131. All the research was conducted in accordance with the Declaration of Helsinki. Written informed consent on behalf of the participants was obtained from their parents or legal guardians.

### 2.3. Data Collection

Data were obtained from the patients’ medical records and the use of questionnaires. They included age, gender, clinical data and comorbidities.

### 2.4. Biological Samples

Nasopharyngeal samples were collected from all the participants in this study (N = 107) using a sterile swab (Plast Labor, Rio de Janeiro, Brazil). The collection methodology was as follows: swabs were introduced into each nostril until there was resistance, then rotated at 180° to acquire the smear. The swabs were stored after collection in 2 mL of viral transport (VTM) and kept under refrigeration (4–8 °C) for a maximum of 24 h until processing. They were then centrifuged at 3584× *g* for 10 min. All the supernatant was stored at −80 °C for later analysis. In addition, tracheal aspirates were collected from all the children with severe CAP (N = 14), using a probe for insertion and tracheal aspiration as deep as possible, discarding the material directly into a collection bottle with a vacuum system (Broncozamm Tr model, Zammi Instrumental Ltd.a, Duque de Caxias, Brazil). All the samples collected by tracheal aspiration were sealed and immediately transported to the Central Public Health Laboratory of Maranhão—LACEN-MA (São Luís, MA, Brazil) for analysis. These samples were then diluted with a mucolytic agent (1% N-acetylcysteine, 1:1) and centrifuged at 1300× *g* for 10 min. The supernatants were stored at −80 °C for further analysis.

### 2.5. Nucleic Acid Extraction

Viral nucleic acid extraction was performed using an Extracta 96 automated system from Loccus company. The genetic material was extracted using the magnetic beads method, with reagents from the MagMax Viral and Pathogen Nucleic Acid Isolation Kit (Thermo Fisher Scientific, Waltham, MA, USA), using 200 μL of the sample, with an elution volume of 60 μL, according to the manufacturer’s instructions.

### 2.6. Molecular Analysis

Viruses were identified by reverse transcription plus real-time quantitative polymerase chain reaction (RT-qPCR) using the following components and given volumes: 1 μL of each primer (200–1000 nM), 0.4 μL of hydrolysis probe ranging from 200 to 400 nM, 12.5 µL of 2 × GoTaq Probe One-Step RT-qPCR master mix (Promega) and 5 µL of cDNA, in a final volume of 25 µL. RT-qPCR analyses were performed in a QuantStudio 6 Flex thermocycler (Thermo Fisher Scientific, Waltham, MA, USA) using the following schedule: one cycle at 95 °C for 10 min, followed by 40 cycles at 95 °C for 10 s each and at 60 °C for 1 min. Positive and negative DNA controls were used in all the RT-qPCR assays performed to avoid false-negative and false-positive results [9].

### 2.7. Statistical Analysis

Data were analyzed using the GraphPad Prism version 6.0 and the Epi-Info version 6.0 software. Categorical variables were compared using χ^2^ or Fisher’s exact test. The statistical significance was defined as *p* < 0.05. Data are expressed as the number of cases and the percentage.

## 3. Results

### 3.1. Clinical and Epidemiological Data

Of the one hundred and seven (107) patients who tested negative for SARS-CoV-2, fourteen (13.1%) had severe CAP and ninety-three (86.9%) had non-severe CAP. No association between CAP severity and a younger age was found. Indeed, among those with severe manifestations of the disease, 75.3% were under 24 months of age. A similar profile was found in patients with non-severe CAP, of whom 71.7% were in the same age range (*p* > 0.005).

The most frequent symptoms amongst the CAP patients were cough (69.1%), fever (63.5%), tiredness (47.6%), expectoration (44.8%), snoring (34.6%), tachypnea (32.7%), runny nose (28.0%) and vomiting (29.0%). Cough (100%), snoring (64.3%), dyspnea (57.1%), nasal obstruction (28.6%) and diarrhea (21.4%) were associated with severe CAP but not with the non-severe form of the disease (*p* < 0.05). Clinical and epidemiological data are depicted in Table 1. There was no significant difference in CAP severity based on gender or comorbidities (*p* > 0.05).

### 3.2. Viral Detection

The tested viruses were detected in 74 (69.2%) of the CAP patients. HRSV was the most prevalent, as 70 out of 107 patients (65.4%) were positive for this virus. HRSV subtype B was detected in 68 patients and HRSV subtype A was detected in 2 patients. In addition, HRV was detected in 4 (3.7%) out of 107 patients. None of the patients were positive for ADV, HMPV, FluA or FluB. RSV infection was associated with a severe respiratory infection, as 13 out of 14 patients (92.9%) with severe ARI were infected with this virus, in comparison to 53 of the 93 non-severe patients (57.0%), which is a significant difference, *p* = 0.015 (Table 2).

### 3.3. Clinical Profile of Children with RSV

Most of the observed clinical symptoms were associated with an RSV infection. The most frequent symptoms in children with RSV were cough (83.3%), fever (74.2%), tiredness (59.1%), expectoration (57.6%), snoring (56.1%), tachypnea (40.9%), vomiting (39.4%), groaning (18.2%) and diarrhea (10.6%). In addition, an association between RSV infection and a younger age (children under 24 months of age) was detected (*p* = 0.012, Table 3).

## 4. Discussion

CAP is the main cause of hospitalization in children, particularly for those under five years of age [3]. In our study, CAP was more frequent in children younger than 24 months, with no correlation with the severity of the infection. This may be related to the variety of viruses that cause ARI in children and the pathogenesis of the viral agents [11]. The analysis of the clinical profile of the patients showed that cough was the main symptom associated with severe ARI, followed by fever, sputum, runny nose and tiredness, in agreement with previous findings [12,13,14]. For example, Li et al. reported that among 2768 patients with ARI, the majority (85.0%) presented cough, corroborating our results [14].

Here, the respiratory syncytial virus was the most prevalent etiological agent, in accordance with studies carried out in other countries that identified it as the most frequent pathogen in children with respiratory infections [15,16,17,18]. In China, a study carried out with 366 children showed a high incidence of RSV when compared to other analyzed viruses [19]. Likewise, Gentile et al. investigated the prevalence of RSV in 6047 Argentine children, reporting that most cases (81.1%) of acute lower respiratory tract infections were caused by RSV [17]. To the best of our knowledge, our study is one of the first to report a high incidence of RSV during the COVID-19 pandemic. On the other hand, Britton and colleagues demonstrated a low frequency of RSV during this period due to the measures taken to prevent the transmission of SARS-CoV-2 [20].

We observed a correlation between a young age and RSV infection. The majority of children with ARI caused by RSV were children younger than 24 months. In this context, Bont et al. showed that young age (children younger than two years) is associated with risk factors for RSV infections, as well as being born during the rainy or cold seasons and exposure to crowds [15]. In our study, RSV was also associated with CAP severity. In fact, this virus is the main etiological agent responsible for hospitalizations and is associated with an increasing severity of viral respiratory infections [21,22]. In reports comparing RSV and SARS-CoV-2 infections, it was found that children with RSV have more severe symptoms, requiring oxygen support, and that this virus can be even more dangerous in the neonatal period, while SARS-CoV-2 infection manifests as a less severe disease [23]. Thus, there is an urgent need for the development of antivirals and vaccines against RSV, given the high incidence and severity of RSV infection in children [24].

RSV infection in children was associated with cough, fever, tiredness, sputum, snoring, tachypnoea, vomiting, groaning and dyspnea. Fever is the main symptom that leads to the search for care from health services; however, it is not an exclusive indicator of ARI, as it occurs in other infections in childhood [25]. Nonetheless, whooping cough is related to RSV infection, coughing being one of the main symptoms reported by primary care providers due to a viral upper-airway infection [12]. Other symptoms such as tiredness, expectoration, snoring, vomiting, moaning and increased respiratory rate may be associated with coughing. Some studies report a correlation between the clinical profile of children with RSV and symptoms such as cough, fever and tachypnoea being the most frequent, corroborating our study [14,26,27].

The high proportion of RSV in children during the first quarter of 2021 was due to a period of higher humidity. In the state of Maranhao, from April 2018 to March 2019, Lopes and collaborators (2020) observed an association between the presence of the virus and the rainy season when analyzing the seasonality of RSV. They observed that RSV circulated from January to May, with a peak in infections in the months of February and March [28]. Similarly, in our previous study, we showed that although RSV was not the most prevalent, it circulated more frequently from January to June in Sao Luis, a city in the northeast of Brazil [9]. In other countries, such as Australia, RSV infection occurs more frequently in the months of April and May [29]. The seasonality of RSV depends on the location and climate; its peak period of infection can vary according to region [30,31,32], but the temporal shift in a RSV epidemic is unlikely to be explained by the emergence of a high-frequency, unreported variant [33].

The rhinovirus was identified at a low incidence in our study, unlike other works that described it as more prevalent [9,34] with a greater presence in adults, mainly in manifestations of coinfection with SARS-CoV-2, as shown in the study by Glass et al. from 2020 [35]. In addition, influenza A and B viruses, adenovirus and metapneumovirus were not identified. This may be due to their low frequency in respiratory infections during the period of the COVID-19 pandemic, as presented in several scientific papers [16,19,36,37,38].

Acute viral infection caused by RSV affects the vast majority of infants younger than 2 years, as shown in our study. The sum of the inflammatory, infectious and immunological alveolar processes results in long-term bronchospasms. Thus, there is great variability of clinical symptomatology in RSV infections [39,40], which can present as upper airway infections (URTI) with cough, runny nose, nasal obstruction and involvement of the lower airways, characterized by bronchiolitis and pneumonia. In our study, we found a low percentage (30%) of wheezing in the infants studied, similar to another study in the Brazilian child population [41], possibly due to the range of symptomatology characteristic of respiratory viruses.

With regard to the prevention of infections caused by RSV in infants, the use of palivizumab has a mainly preventative role, especially in preterm infants with a gestational age < 35 weeks, considerably reducing hospitalizations due to RSV, as shown in the study by Blanken et al. from 2020 [42].

Taking into account the benefits of the prophylactic use of palivizumab in the prevention of RSV infections in premature infants, use of this drug in the Unified Health System (SUS) of Brazil was approved in November 2012 by Recommendation Report no. 16, by the National Commission for the Incorporation of Technologies in the SUS (Conitec) [43]. The protocol for the use of palivizumab to prevent RSV infection was updated in 2018 by Joint Ordinance SCTIE/SAS No. 23, 3 October 2018, and by Technical Note No. 45/2019-CGAFME/DAF/SCTIE/MS [44]. Prophylaxis with palivizumab is used only during RSV season, and is offered free of charge by the SUS for premature children born at a gestational age of less than or equal to 28 weeks, and aged less than 1 year. Children older than 1 year but younger than 2 years with a chronic lung disease such as bronchopulmonary dysplasia, or a congenital heart disease with demonstrated hemodynamic repercussions, are also included in the prophylactic treatment [45].

## 5. Conclusions

In conclusion, we verified that RSV was the main virus detected in children with CAP and also associated with severe CAP, showing the importance of the initiative for the development of antivirals and vaccines to treat and prevent RSV infections, especially for children younger than 24 months of age, who are more consistently affected by RSV. In addition, we highlighted the importance of prioritizing the molecular diagnosis of respiratory viruses, which directly contributes to better medical interventions, promoting a faster and more specific treatment, and avoids the possible misuse of medicines, for example, the use of antibiotics to treat infections of a viral rather than bacterial origin.

## Figures and Tables

**Figure 1 viruses-15-01306-f001:**
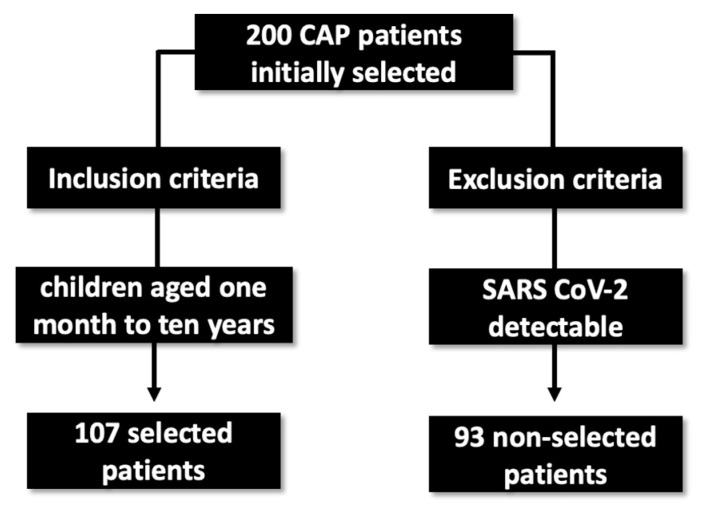
Sample selection flowchart for the research.

**Table 1 viruses-15-01306-t001:** Frequency of community-acquired pneumonia in association with severity.

Variables	CAP (n = 107)	Severe CAP (n = 14)	Non-Severe CAP (n = 93)	*p* *
Age (>24 months)	80 (74.8%)	10 (71.4%)	70 (75.3%)	0.999
Male gender	53 (49.5%)	7 (50.0%)	46 (49.5%)	0.999
Cough	74 (69.1%)	14 (100.0%)	60 (64.5%)	0.004 *
Fever	68 (63.5%)	11 (78.6%)	57 (61.3%)	0.249
Tiredness	51 (47.6%)	10 (71.4%)	41 (44.1%)	0.083
Expectoration	48 (44.8%)	9 (64.3%)	39 (41.9%)	0.758
Snoring	37 (34.6%)	9 (64.3%)	28 (30.1%)	0.017 *
Dyspnea	49 (45.8%)	9 (64.3%)	40 (43.0%)	0.159
Tachypnea	35 (32.7%)	7 (50.0%)	28 (30.1%)	0.219
Runny nose	30 (28.0%)	5 (35.7%)	25 (26.9%)	0.529
Vomiting	31 (29.0%)	4 (28.6%)	27 (29.0%)	0.999
Nasal obstruction	7 (6.5%)	4 (28.6%)	3 (3.2%)	0.005 *
Diarrhea	7 (6.5%)	3 (21.4%)	4 (4.3%)	0.046 *
Wheezing	28 (26.2%)	2 (14.3%)	26 (28.0%)	0.347
Cyanosis	7 (6.5%)	2 (14.3%)	5 (5.4%)	0.227
Abdominal pain	6 (5.6%)	2 (14.3%)	4 (4.3%)	0.175
Congenital heart disease	8 (7.5%)	1 (7.1%)	7 (7.5%)	0.999
Hematological diseases	1 (0.9%)	1 (7.1%)	0 (0.0%)	0.13
Prematurity	12 (11.2%)	0 (0.0%)	12 (12.9%)	0.359
Neurological diseases	1 (0.9%)	1 (7.1%)	0 (0.0%)	0.13
Chronic kidney disease	3 (2.8%)	0 (0.0%)	3 (3.2%)	0.999
Chronic liver disease	4 (3.7%)	0 (0.0%)	4 (4.3%)	0.999
Asthma	1 (0.9%)	0 (0.0%)	1 (1.1%)	0.999
Visceral leishmaniasis	4 (3.7%)	1 (7.1%)	3 (3.2%)	0.434
Pulmonary tuberculosis	2 (1.9%)	0 (0.0%)	2 (2.1%)	0.999
HIV infection	1 (0.9%)	0 (0.0%)	1 (1.1%)	0.999
Mesenteric cyst	1 (0.9%)	0 (0.0%)	1 (1.1%)	0.999
Absence of comorbidity	62 (57.9%)	10 (71.4%)	52 (55.9%)	0.386

Note: Values are shown as the number of individuals and the percentage (%) in parentheses. Abbreviations: CAP, community-acquired pneumonia. * Significant differences between the severe and non-severe CAP groups were calculated by using Fisher’s exact tests.

**Table 2 viruses-15-01306-t002:** Frequency of the viruses detected by qPCR in 107 children with CAP according to CAP severity.

Virus Positivity	CAP, n = 107	Severe CAP, n = 14	Non-Severe IRA, n = 93	*p* *
RSV	70 (65.4)	13 (92.8%)	57 (61.2%)	0.031
HVSR A	2 (1.9%)	0 (0.0%)	2 (2.1%)	1
HVSR B	68 (63.5%)	13 (92.8%)	55 (59.1%)	0.016
Rhinovirus	4 (3.7%)	1 (7.1%)	3 (3.2%)	0.434
FluA	ND	ND	ND	--
FluB	ND	ND	ND	--
HADV	ND	ND	ND	--
HMPV	ND	ND	ND	--
HCoV 229E	7 (6.5%)	3 (21.4%)	4 (4.3%)	0.046
HCoV NL63	ND	ND	ND	--
HCoV OC43	ND	ND	ND	--
HCoV HKU1	ND	ND	ND	--
HPIV 1	ND	ND	ND	--
HPIV 2	ND	ND	ND	--
HPIV 3	ND	ND	ND	--
Coinfections	7 (6.5%)	3 (21.4%)	4 (4.3%)	0.046
No virus detected	33 (30.8%)	0 (0.0%)	33 (35.5%)	1.000

Note: The n values of individuals and the percentage (%) in parentheses. Abbreviations: ARI, acute respiratory infection; RSV, respiratory syncytial virus; HRV, human rhinovirus; FluA, influenza A virus; FluB, influenza B virus; ADV, adenovirus; HMPV, human metapneumovirus; qPCR, real-time quantitative polymerase chain reaction; ND, not detected. * Significant differences between the severe and non-severe CAP groups were calculated using Fisher’s exact tests.

**Table 3 viruses-15-01306-t003:** Clinical profile of children testing positive or negative for RSV according to CAP severity.

Variables	Total CAP (n = 107)	VSR Detected (n = 66)	VSR Not Detected (n = 41)	*p* *
Young age, %	80 (74.7%)	55 (83.3%)	25 (60.9%)	0.012 *
Cough	74 (69.1%)	55 (83.3%)	19 (46.3%)	<0.0001 *
Fever	68 (63.5%)	49 (74.2%)	19 (46.3%)	0.006 *
Tiredness	51 (47.6%)	39 (59.1%)	12 (29.3%)	0.003 *
Expectoration	48 (44.8%)	38 (57.6%)	10 (24.4%)	<0.001 *
Snoring	37 (34.6%)	37 (56.1%)	0 (0.0%)	<0.0001 *
Dyspnea	49 (45.8%)	28 (42.4%)	21 (51.2%)	0.427
Tachypnoea	35 (32.7%)	27 (40.9%)	8 (19.5%)	0.033 *
Runny nose	30 (28.0%)	21 (31.8%)	9 (21.9%)	0.376
Vomiting	31 (29.0%)	26 (39.4%)	5 (12.2%)	0.003 *
Nasal obstruction	7 (6.5%)	6 (9.1%)	1 (2.4%)	0.246
Diarrhea	7 (6.5%)	7 (10.6%)	0 (0.0%)	0.042 *
Wheezing	28 (26.2%)	20 (30.3%)	8 (19.5%)	0.262
Cyanosis	7 (6.5%)	4 (6.1%)	3 (7.3%)	>0.999
Abdominal pain	6 (5.6%)	3 (4.5%)	3 (7.3%)	0.673
Congenital heart disease	12 (11.2%)	7 (10.6%)	5 (12.2%)	>0.999
Hematological diseases	13 (12.1%)	10 (15.1%)	3 (7.3%)	0.362
Prematurity	12 (11.2%)	12 (18.2%)	0 (0.0%)	0.003 *
Neurological diseases	8 (7.5%)	1 (1.5%)	7 (17.1%)	0.004 *
Cough	2 (1.9%)	0 (0.0%)	2 (4.9%)	0.144
Fever	12 (11.2%)	7 (10.6%)	5 (12.2%)	>0.999
Tiredness	1 (0.9%)	0 (0.0%)	1 (2.4%)	0.383
Chronic kidney disease	3 (2.8%)	1 (1.5%)	2 (4.9%)	0.556
Chronic liver disease	4 (3.7%)	0 (0.0%)	3 (7.3%)	0.144
Asthma	1 (0.9%)	0 (0.0%)	1 (2.4%)	0.383
Visceral leishmaniasis	4 (3.7%)	1 (1.5%)	3 (7.3%)	0.156
Pulmonary tuberculosis	2 (1.9%)	0 (0.0%)	2 (4.9%)	0.144
Mesenteric cyst	1 (0.9%)	0 (0.0%)	1 (2.4%)	0.383
Absence of comorbidity	62 (57.9%)	51 (77.2%)	11 (26.9%)	<0.0001 *

Note: Values are shown as the number of individuals and the percentage in parentheses. * Significant differences between the severe and non-severe CAP groups were calculated using Fisher’s exact tests for the groups infected with RSV and the groups infected by other pathogens. Young age was considered to be patients under 24 months.

## Data Availability

Data are available and will be sent by email upon request.

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
