# Peer review of "High Incidence of Respiratory Syncytial Virus in Children with Community-Acquired Pneumonia from a City in the Brazilian Pre-Amazon Region"

_viruses, 2023, doi:10.3390/v15061306_

Round 1
Reviewer 1 Report
The study investigated, the incidence of respiratory syncytial virus (RSV) as well its subtypes (RSV A and B), adenovirus (ADV), rhinovirus (HRV), metapneumovirus (HMPV), coronavirus (NL63, OC43, 229E, HKU1), parainfluenzavirus subtypes (PI1, PI2, and PI3), Bocavirus and influenza A and B viruses (FluA and FluB) in children diagnosed with CAP during the COVID-19 pandemic. The 200 clinically confirmed children with CAP were recruited and 107 qPCR-negative patients for SARS-CoV-2 were included in the study. The viruses were identified in 69.2% of patients. RSV infections were the most frequently identified (65.4%) being its type RSV B the most prevalent (63.5%). In addition, HCoV 229E and HRV were detected in 6.5% and 3.7% of the patients, respectively. RSV type B was associated with severe ARI and low age (less than 24 months).
Comments and Suggestions
Abstract
The phase on lines 26, and 27 is not clear and does not have the meaning that the authors want to inform.
Methodology
Of the evaluated samples, how many swabs were obtained, and how many nasopharyngeal aspirates? What was the difference in the treatment of the samples? Are these samples added together or not?
The result
I did not find figure one cited in the text.
In conclusion
In conclusion, the authors could also talk about the importance of identifying the viral types and that making this identification a routine would be important to minimize the effects of ineffective treatments that are often used.
Author Response
REPLY LETTER TO EVALUATORS
Hello dears, all parts of the text that reported similarities with other works have been updated as requested.
All changed parts of the text are highlighted in green.
Regarding the arguments of the evaluators, follow the notes below:
EVALUATOR 1:
- Summary updated as requested;
- With regard to information on the methodology: nasopharyngeal samples were collected from all participants in this study (N=107). The swabs were stored after collection in 2 mL of viral transport (VTM), and these were kept under refrigeration (4 - 8ºC) for a maximum of 24 hours until processing. All supernatant was stored at -80°C for later analysis. In addition, tracheal aspirates were collected from all children with severe CAP (N=14). All samples collected by tracheal aspirate were sealed and immediately transported for analysis. These samples were then diluted with a mucolytic agent (1% N-acetylcysteine, 1:1) and centrifuged at 1,300 x g for 10 minutes. Supernatants were stored at -80ºC for further analysis. This whole paragraph was reworded and added in the work;
- As requested, the entire work completion topic has been updated.

Reviewer 2 Report
In my opinion, the manuscript is of interest and can be published after some revision. The Conclusions are generally correct, but, unfortunately, the articles does not contain data on the effectiveness of RSV infection prevention and treatment. In this regard, I think, it is necessary to draw conclusions closer to the results obtained.
The manuscript needs to be more carefully revised. For example, authors need to enter the correct title for Table 2. In addition, in the tables and text, it is necessary to follow the generally accepted abbreviation more exactly (RSV instead VRS) etc.
Author Response
REPLY LETTER TO EVALUATORS
Hello dears, all parts of the text that reported similarities with other works have been updated as requested.
All changed parts of the text are highlighted in green.
Regarding the arguments of the evaluators, follow the notes below:
EVALUATOR 2:
- As per your suggestion, the conclusions have been updated and the final text has been revised;

Reviewer 3 Report
Estimated Authors of the paper "High incidence of respiratory syncytial virus in children with community acquired pneumonia from a city in the Brazilian Pre-Amazon region",
I've read with great interest your paper dealing with a both interesting and significant topic (i.e. occurrence of RSV infections). However, despite the potential significance even for the general practitioners, I think that several improvements will be required before the full acceptance of this study. More precisely:
1) To begin with, a conceptual concern: according to the introduction, the present paper has been envisaged in order to describe the occurrence of RSV infections in the targeted population of CAP (i.e. Therefore herein, we aimed at investigating the incidence of RSV as well as of its subtypes (RSV A and B), ADV, HRV, HMPV, coronavirus (NL63, OC43, 229E, HKU1), parainfluenzavirus subtypes (PI1, PI2 and PI3), bocavirus and influenza A and B (FluA and FluB) viruses in children with ARI), but the discussion is a little bit too generically designed, and should be refocused (according to the reporting of data, see Table 2) on RSV and its characteristics.
2) Reporting of results should be a little bit refocused: data from Figure 1 could be included in Table 1, making it consistent with the following Table 2; such approach would also allow the reader to clearly understand whether age would represent or not a significant risk factor for severe CAP
3) In your study, distribution has been assessed through an univariate analysis: have you considered a possible multivariable approach in order to weight in the potential role of covariates represented by demographic factors on the development of signs and symptoms?
4) discussion should consider that several preventive options have been and will be made available to the healthcare providers, including mAb (e.g. palivizumab, nirsevimab) and vaccines: assessing whether risk factors such as malformations, pre-existing respiratory disorders and heart diseases are or not truly associated with increase risk for RSV infections is of absolute relevance for future preventive strategies.
5) Similarly, Authors should discuss some specificities of their results, including the very low rate of wheezing among the cases of RSV compared to other cases.
6) Authors should discuss about the very high drop out rate associated with diagnosis of SARS-CoV-2 infection, and the eventual low sampling size.
Author Response
REPLY LETTER TO EVALUATORS
Hello dears, all parts of the text that reported similarities with other works have been updated as requested.
All changed parts of the text are highlighted in green.
Regarding the arguments of the evaluators, follow the notes below:
EVALUATOR 3:
- 1) More information was added in the discussion according to the results obtained in the study;
- 2) As suggested, the data in Figure 1 has been relocated and included in Table 1;
- 3) We appreciate the arguments, but we did not perform the multivariate analyses. However, we will take these considerations into future work.
- 4) Data on preventive methods for RSV infection have been added to the discussion;
- 5) As requested, more direct data on the wheezing rate was added to the discussion;
- 6) With regard to the dropout rate, this is actually the direction of the study according to the inclusion and exclusion criteria, as in this study we excluded all patients who had a detectable result for SARS-CoV-2, which were destined for another research from our same group.

Round 2
Reviewer 3 Report
Estimated Authors,
I've appreciated the efforts you paid in order to improve the overall quality of your paper. I could say that I'm extensively satisfied by your work, but (alas!) some minor improvements are still required. But I'm quite confident that all my concerns (please note that all of them are minor ones!) could be faced in a very short timeframe.
1) Authors have initially sampled 200 patients; it would be interesting to explain 200 patients: have you deliberately chosen to sample 200 consecutive patients assessed by your clinics? please explain.
2) It seems that of these 200 patients, 93 were then excluded from the analyses as SARS-CoV-2 positive. A simple flow-chart would be particularly useful in order to explain the inclusion process.
3) As a supplementary information: did you identify any cross-infection SARS-CoV-2 + RSV?
There are several minor typos scattered across the text, but I think that a simple double check could remove all of them (e.g. row 179, "NOTE" has been removed from the text)
4) please explain in the main captions of the tables whether the p value is referred to a Chi squared or a Fisher's exact test.
Author Response
REPLY TO THE COMENTS OF THE REVIEWER
Reviewer #3,
All changed parts of the text are highlighted in green.
Regarding the arguments of the evaluators, follow the notes below:
- Authors have initially sampled 200 patients; it would be interesting to explain 200 patients: have you deliberately chosen to sample 200 consecutive patients assessed by your clinics? please explain.
We thank the referee for this comment. Please note that we have now added more information in study population (page 2, in line 82).
- It seems that of these 200 patients, 93 were then excluded from the analyses as SARS-CoV-2 positive. A simple flow-chart would be particularly useful in order to explain the inclusion process.
We thank the referee for this comment. Please note that we have now included a flow-chart (Figure 1, page 3).
- As a supplementary information: did you identify any cross-infection SARS-CoV-2 + RSV?
There are several minor typos scattered across the text, but I think that a simple double check could remove all of them (e.g. row 179, "NOTE" has been removed from the text)
We thank the referee for this comment. We agree that identify cross-infection SARS-CoV-2 + RSV would give a better information about respiratory infection and, therefore, futures studies will be performed using this analysis. We did not identify cross-infection with SARS-CoV-2 because according to our exclusion criteria, patients detectable for SARS-CoV-2 were not included in this research.
- Please explain in the main captions of the tables whether the p value is referred to a Chi squared or a Fisher's exact test.
We thank the referee for this comment. As requested, data on the statistical methods used were added to the table captions. Therefore, the p values ​​of the analyzes were evaluated by Fisher's exact test.
